# Oxidation-reduction and photophysical properties of isomeric forms of Safranin

**Eskil M. E. Andersen**[1], **Hsin Wang**[2], **Joshua S. H. Khoo**[3], **Jose F. Cerda**[4], **Ronald L. Koder**[3,5]*

**1** Department of Biochemistry, The City College of New York, New York, NY, United States of America, **2** Department of Chemistry, The City College of New York, New York, NY, United States of America, **3** Department of Physics, The City College of New York, New York, NY, United States of America, **4** Department of Chemistry, St. Joseph's University, Philadelphia, PA, United States of America, **5** Graduate Programs of Physics, Biology, Chemistry and Biochemistry, The Graduate Center of CUNY, New York, NY, United States of America

* rkoder@ccny.cuny.edu

**Data Availability Statement:** All relevant data are within the paper and its Supporting Information files.

**Funding:** This work was supported by a Grant from the New York State Energy Research and

## Abstract

Safranine O is widely used in the bioenergetics community as an indicator dye to determine membrane potentials and as an electron transfer mediator in potentiometric titrations. Here we show that two different commercial preparations of Safranine O contain less than sixty percent by weight of the title compound, with the rest primarily consisting of two closely related safranine isomers. All three major isomer components were isolated using reverse phase HPLC and their structures determined using mass spectrometry and two-dimensional NMR. These Safranines have two-electron midpoint potentials ranging from −272 to −315 mV vs. SHE. We have also investigated the absorption and fluorescence spectra of the compounds and found that they display distinct spectral and photophysical properties. While this mixture may aid in Safranine O's utility as a mediator compound, membrane potential measurements must take this range of dye potentials into account.

## Introduction

In order to fully understand biological electron transfer processes and the reactions coupled to them in metabolism, it is necessary to determine the reduction potentials of respiratory and photosynthetic electron transfer cofactors and side chains. These protein properties are typically investigated in solution, necessitating the transport of electrons through the solvent to react with the dissolved protein analyte. Biological redox couples are commonly inert towards electrode surfaces, and the major reason for this is that the redox centers tend to be shielded by the protein structure which prevents close contact with the electrode surface. Oxidation-*reduction mediators*, which are small organic molecules are therefore required during solution potentiometric measurements to transfer electrons between the electrode and the analyte. Mediators also have the added benefit of creating electrical poise during measurements which helps the experimentalist alter the potential of the solution in a controlled manner.

Safranine (also known as Safranine O, Safranin, Safranine T, or basic red is often used as an electron transfer mediator in potentiometric measurements and as an internal colorimetric

Development Authority (NYSERDA grant #NYS115213 to RLK). NIH grants S10OD018509, S10OD016432, and P41GM066354. URL: https://www.nyserda.ny.gov/ The funders had no role in study design, data collection and analysis, decision to publish, or preparation of the manuscript.

**Competing interests:** The authors have declared that no competing interests exist.

indicator in reduction potential determinations both in solution [1, 2] and in membranes [3–5]. It is a synthetic phenazine derivative that may be prepared from reacting a 1:1 mixture of o-toluidine and 1.4-diamino-2-methylbenzene followed by nucleophilic addition of aniline [6]. The compound has its origins in the burgeoning dye industry of the 19th century and was first isolated in 1859 [7]. The reduction potential of Safranine O at pH 7.0 was first reported by Prince *et al*. in 1981 to be −280 mV *vs* the Standard Hydrogen Electrode (SHE) as determined by polarography [8]. However, based on our findings this value may have been an average of multiple safranine isomers.

In addition to being useful as an electron transfer mediator Safranine has received considerable interest because of its photoreduction properties [9–11]. It has been demonstrated to have applications in a number of areas such as solar energy conversion [12, 13], analytical chemistry [14, 15], as a histological stain [16, 17], and in synthetic organic chemistry as an initiator of photopolymerization [18–20]. Many natural phenazine derivatives are biosynthesized by bacteria as well where they play roles in processes such as electron shuttling to terminal acceptors, modification of cellular redox states, regulation of gene expression, and biofilm formation [21]. Finally, we have synthesized a series of safranine analogues for use as artificial electron transfer cofactors in *de novo* designed enzymes [22, 23].

In the scientific literature commercial sources of Safranine O are assumed to consist of a homogeneous compound. We have discovered that they in fact contain a heterogenous mixture of isomeric safranines that are resistant to separation by normal phase chromatography. We have therefore separated and isolated the three major components of a commercial sample of Safranine using C18 reverse phase HPLC. These three components make up more than 90% of the total safranine content in the mixture. We then determined their key physicochemical properties, most importantly their reduction potentials, as well as their ultraviolet and visible absorption and fluorescence properties. Finally, we determined the chemical structures of the components using two-dimensional NMR spectroscopy.

## Materials and methods

### Chemicals

Safranine O samples were purchased from Sigma-Aldrich (St. Louis, MO) and Santa Cruz Biotech (Dallas, TX). HPLC grade Acetonitrile was purchased from *P212121* (Ann Arbor, MI) and HPLC grade trifluoroacetic acid was purchased from VWR (Radnor, PA). The water used in all purifications and experiments was obtained from a MilliQ system and had a minimum resistivity of 18 MΩ. The mediators used in the potentiometric measurements were methyl viologen (1,1′-Dimethyl-4,4′-bipyridinium dichloride) and pyocyanin (5-Methylphenazin-1-one) also purchased from Sigma-Aldrich. Sodium hydrosulfite and potassium ferricyanide for potentiometric experiments were purchased from VWR.

### Chemical purification

Chromatography was performed on a Shimadzu (Kyoto, Japan) HPLC equipped with 2 LP-6AD pumps coupled to an SPD-M20A PDA spectrophotometer. The HPLC column used was a 20x250 mm *Proto 300* C18 with 10 μm particle size from Higgins Analytical (Mountain View, CA). Safranine O was dissolved in 4:1 water: acetonitrile. The mixture was separated using an isocratic flow of 27% acetonitrile, 73% water running at a flow-rate of 18 mL/min. After separation, safranine samples were rotary evaporated to remove acetonitrile, and the remaining water was removed by lyophilizing the frozen analytes overnight in 50 mL falcon tubes. During lyophilization, analytes were protected from light by aluminum foil wrappings.

## Absorption and fluorescence spectroscopy

The buffer used in all spectroscopy and potentiometry measurements consisted of 50 mM sodium phosphate adjusted to pH 7.0 with phosphoric acid or sodium hydroxide. Absorption spectra were collected on a Hewlett-Packard (Palo Alto, CA) HP 8452A diode array spectrophotometer equipped with OLIS GlobalWorks software for data collection and analysis. Fluorescence spectra were collected on an OLIS (Athens, GA) DM-45 fluorimeter. All Safranines were excited at their wavelength of maximal absorption. Samples were diluted to an absorption of 0.1 AU and degassed with nitrogen in a 1x1 (width x depth) cm quartz cuvette before measurements.

## Structural determination

NMR spectra were recorded in perdeuterated DMSO on a Bruker (Billerica, MA) Avance III spectrometer operating at 600 MHz for $^1$H, 150 MHz for $^{13}$C, and 60 MHz for $^{15}$N. Assignments were confirmed with $^1$H-$^{13}$C HSQC and HMBC, $^1$H-$^{15}$N HMBC, and $^1$H-$^{13}$C 1, 1-ADEQUATE experiments [24–26]. Amberized NMR tubes were used to protect the samples from photodegradation.

## Reduction potential determination

Solution potentiometry [27, 28] was used to determine the redox potentials of the Safranines. Briefly: an airtight cuvette is fitted with a platinum measuring electrode and an Ag/AgCl saturated potassium chloride reference electrode. The reference electrode was standardized using hydroquinone at pH 7 and at 25°C [29]. Potentials were determined in 50 mM sodium phosphate solution pH 7.0 using 5 μM methyl viologen and 5 μM pyocyanin as electron mediators and to establish poise during measurements.

Samples were degassed via rubber septum-sealed ports with nitrogen scrubbed of oxygen using a zinc-vanadium bubbler oxygen scrubbing system [30]. Samples were reduced using microliter additions of sodium dithionite in a sodium hydroxide solution (pH>10) and their absorption spectra were determined after each addition. Once the samples had been fully reduced, they were oxidized stepwise using microliter additions of a potassium ferricyanide solution in water while measuring the absorption spectra going in the oxidizing direction to measure the degree of hysteresis. The changes in absorption at the peak reduced wavelength of each safranine was measured and fitted using the Nernst equation (Eq 1) with the number of electrons (n) fixed at 2. In this equation $E_{cell}$ is the potential of the cell relative to the SHE ($E_{SHE}^\theta$), R is the universal gas constant, T is the temperature in Kelvin, F is Faraday's constant, and [ox] and [red] are the concentrations of oxidized and reduced safranine, respectively.

$$E_{cell} = E_{SHE}^\theta - \frac{RT}{nF} ln \frac{[ox]}{[red]} \qquad (1)$$

# Results and discussion

## Purification

Separation on an analytical C18 HPLC column revealed that the Safranine O mixture is heterogenous and contains a mixture of several components with Safranine type spectra (*Fig 1*). Of these compounds **I**, **II** and **III** constitute > 95% of the total area and we therefore decided to investigate the properties of these compounds further. At least 20 mg of each of compound was successfully isolated by RP-HPLC for NMR structure determination.

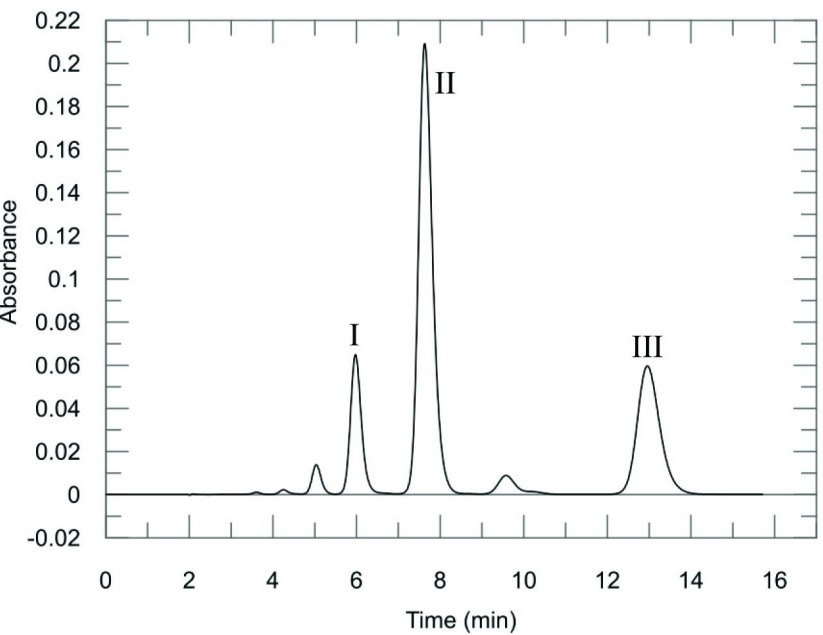

**Fig 1. Chromatogram showing the separated peaks of the commercial Safranine O mixture.** The mixture was separated on a Higgins Proto 300 4.6x250 mm column. The chromatographic method is isocratic and consists of 27% acetonitrile/73% water with 0.1% trifluoroacetic acid at a flowrate of 2 mL/min. The chromatographic trace was made by plotting the absorption at 520 nm vs. time. The peaks are numbered in order of elution and correspond to the scheme in Table 1.

## Absorption and fluorescence spectroscopy

As shown in Fig 2 and Table 1; safranine **I** exhibits absorption peaks at 206, 278 and 524 nm in the experimental buffer. Contrary to **II** and **III**, **I** is not fluorescent. **II** shows absorption peaks at 206, 250, 276 and 520 nm and has an emission maximum at 581 nm. **III** evinces absorption maxima at 210, 250, 278 and 516 nm with an emission maximum at 594 nm. Not only does **III** absorb at a shorter wavelength (516 nm), but it also emits at a longer wavelength (594 nm). This implies that there must be a greater level of vibrational relaxation in this molecule as compared to **II**. In addition, the quantum yield for **II** is greater, and for the same absorption of 0.1 AU, **II** is more than 3 times as fluorescent as **III**.

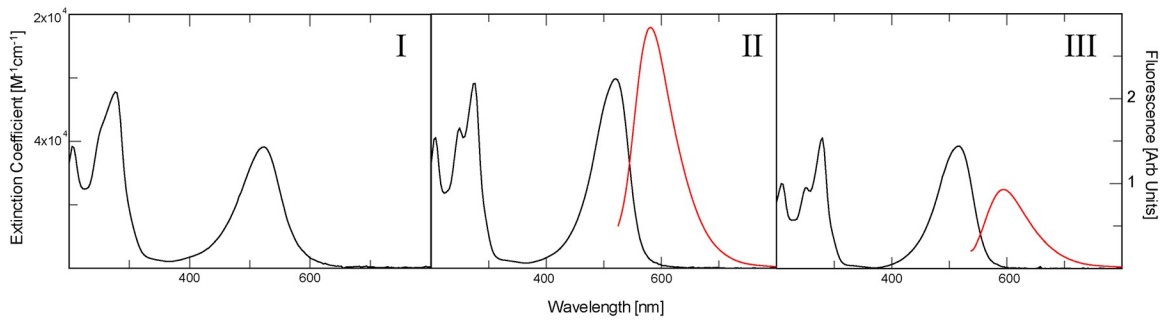

**Fig 2. Absorption and fluorescence spectra of three safranine isomers.** The absorption extinction coefficient is shown in black, while fluorescence is shown in red. I) The absorbance spectra of safranine **I** (this compound is not fluorescent). II) The absorption and fluorescence spectra of safranine **II**. III) The absorption and fluorescence spectra of safranine **III**. Fluorescence spectra were obtained by exciting the Safranine at their absorption maxima around 520 nm. The absorption at the wavelength of excitation was set to 0.1 AU at the long wavelength absorption maximum for both emission spectra.

**Table 1. Safranine O components and their properties.**

| Compound | Compound | Redox potential [mV vs. SHE] | Peak wavelengths [nm] | Extinction coefficient [$M^{-1}cm^{-1}$] | Emission $\lambda_{max}$[nm] | Relative maximal emission |
|---|---|---|---|---|---|---|
| 2,8-diamino-3,9-dimethyl-10-phenylphenazin-10-ium | I | −315 ± 2 | 206 | 19,376 | none detected | N/A |
| | | | 278 | 27,833 | | |
| | | | 524 | 19,107 | | |
| 2,8-diamino-3,7-dimethyl-10-phenylphenazin-10-ium | II | −272 ± 2 | 206 | 20,644 | 581 | 3.06 |
| | | | 250 | 22,115 | | |
| | | | 276 | 29,372 | | |
| | | | 520 | 29,800 | | |
| 2,8-diamino-3,6-dimethyl-10-phenylphenazin-10-ium | III | −276 ± 5 | 210 | 13,437 | 594 | 1 |
| | | | 250 | 12,743 | | |
| | | | 278 | 20,579 | | |
| | | | 516 | 19,209 | | |

## Structural determination

All three safranines have identical masses as determined using mass spectrometry. The major HPLC component, **II**, displays [1]H and [13]C-spectra characteristic of a molecule with C2 symmetry (Fig 3), and can be readily identified as authentic safranine O. The assignments of all resonances of **II** are straightforward with the use of two-dimensional HSQC [31] and HMBC [32] ([13]C as well as [15]N) spectroscopies.

The other two HPLC peak compounds (**I** and **III**) do not have such symmetry properties. We posited initially that **IV** in Fig 4 might be a good candidate for peak #5 because its [1]H-spectrum displays 4 aromatic ring protons that are singlet or weakly-coupled. None of the other NMR spectra are consistent with this picture. An unambiguous clue was given by the 1,1-ADEQUATE [26] experiment which showed that one of the two $NH_2$-bearing aromatic carbons at ~156 ppm is adjacent to two CH-neighbors (boxed peaks in Fig 5). This is only possible if on one of the side rings the $NH_2$ and $CH_3$ groups are not adjacent to each other. **III** is thus

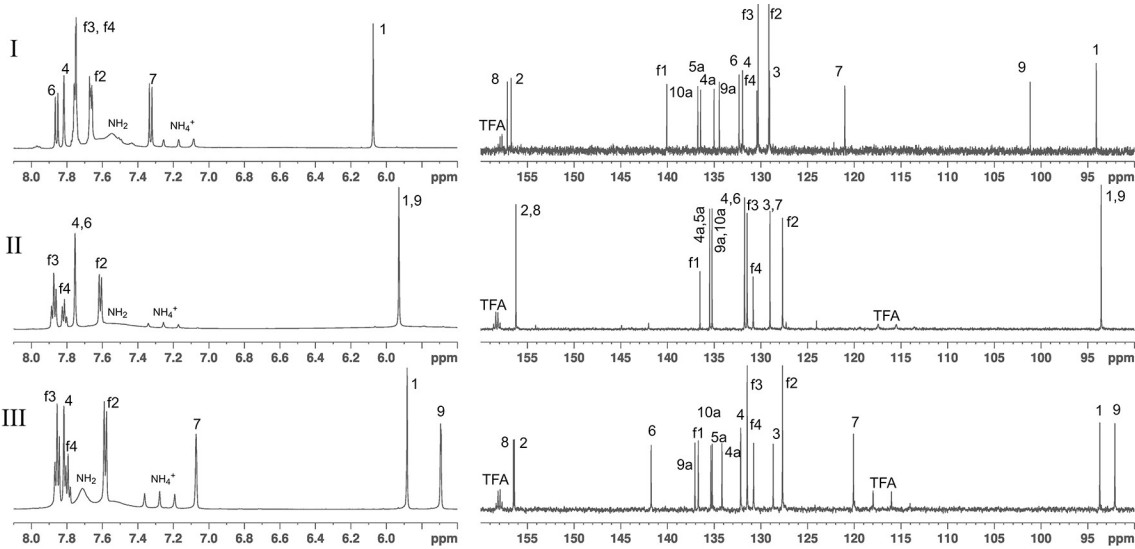

**Fig 3.** One-dimensional [1]H (left) and [13]C (right) spectra and resonance assignments, exclusive of the methyl region, in DMSO-d6 of safranines I-III.

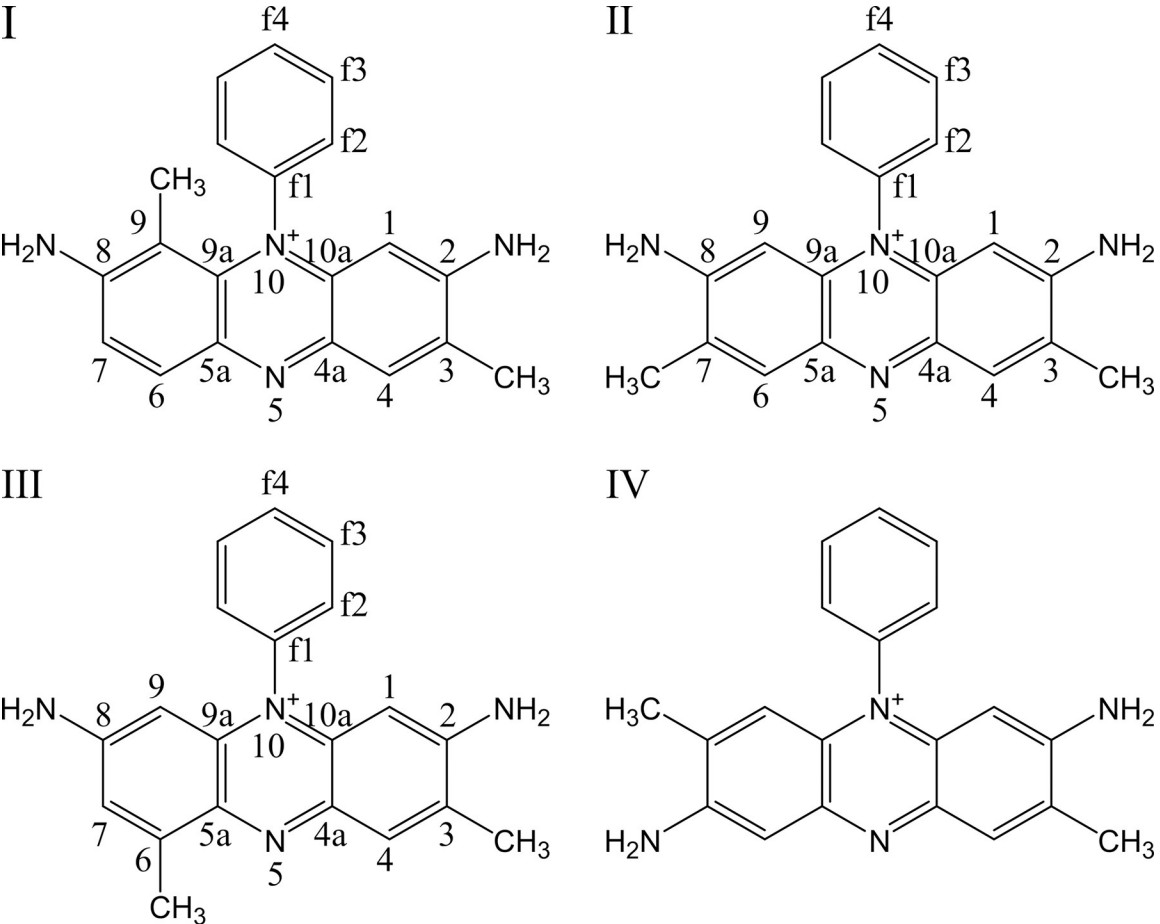

**Fig 4. The three major Safranine structures determined by NMR.** The number annotations in red refer to the atom assignments done during the analysis of the NMR spectra and follow the biochemical numbering system of flavin and flavin-like molecules. The IUPAC names of the compounds: (**I**) 2,8-diamino-3,9-dimethyl-10-phenylphenazin-10-ium. (**II**) 2,8-diamino-3,7-dimethyl-10-phenylphenazin-10-ium (**III**) 2,8-diamino-3,6-dimethyl-10-phenylphenazin-10-ium (**IV**) 2,7-diamino-3,8-dimethyl-10-phenylphenazin-10-ium–an alternate possible structure of III ruled out on the basis of the 1,1 ADEQUATE spectrum depicted in *Fig 5*.

the only remaining candidate. Indeed with this insight, all NMR data fall into place: 1) the carbon resonances can be uniquely assigned with 1, 1-ADEQUATE and agrees completely with [13]C-HMBC; 2) the weak ~2 Hz splitting of $H$(9) is the result of a coupling with the meta-positioned $H$(7) whereas the broad feature of $H$(7) resulted from additional 4-bond J-coupling with the $CH_3$ [$H$(18)]; 3) the information-rich [15]N-HMBC data (Fig 6) can be completely understood with 3-bond, 2-bond, and even a few weak 4-bond J-couplings. With the discovery that one methyl group rearranges itself on the ring in **III**, **I** can be readily identified to be structured as in Fig 4. The ortho-positioned $H$(6) and $H$(7) displayed a J-coupling of 9.1 Hz with each other. All the other NMR data including 1,1-ADEQUATE are consistent with this model and can be interpreted in a straightforward manner. Interestingly, the phenyl ring protons $H$(f2), $H$(f3), $H$(f4) do not display the expected doublet, triplet, and triplet patterns, respectively, as in the other two compounds. In this case $H$(f3) and $H$(f4) are nearly degenerate and strong-interaction of J-couplings result in complicated multiplet patterns.

Also, there is a 1:1:1 triplet in the region of 7.0–7.4 ppm in [1]H-spectra (Fig 3) with a splitting of 50.8 Hz due to J-coupling to the quadrupolar [14]N. The [15]N-HMBC (Fig 6) on the other hand showed a [1]H – [15]N splitting of 70 Hz at 23.0 ppm. These are characteristic of $NH_4^+$ ion [33].

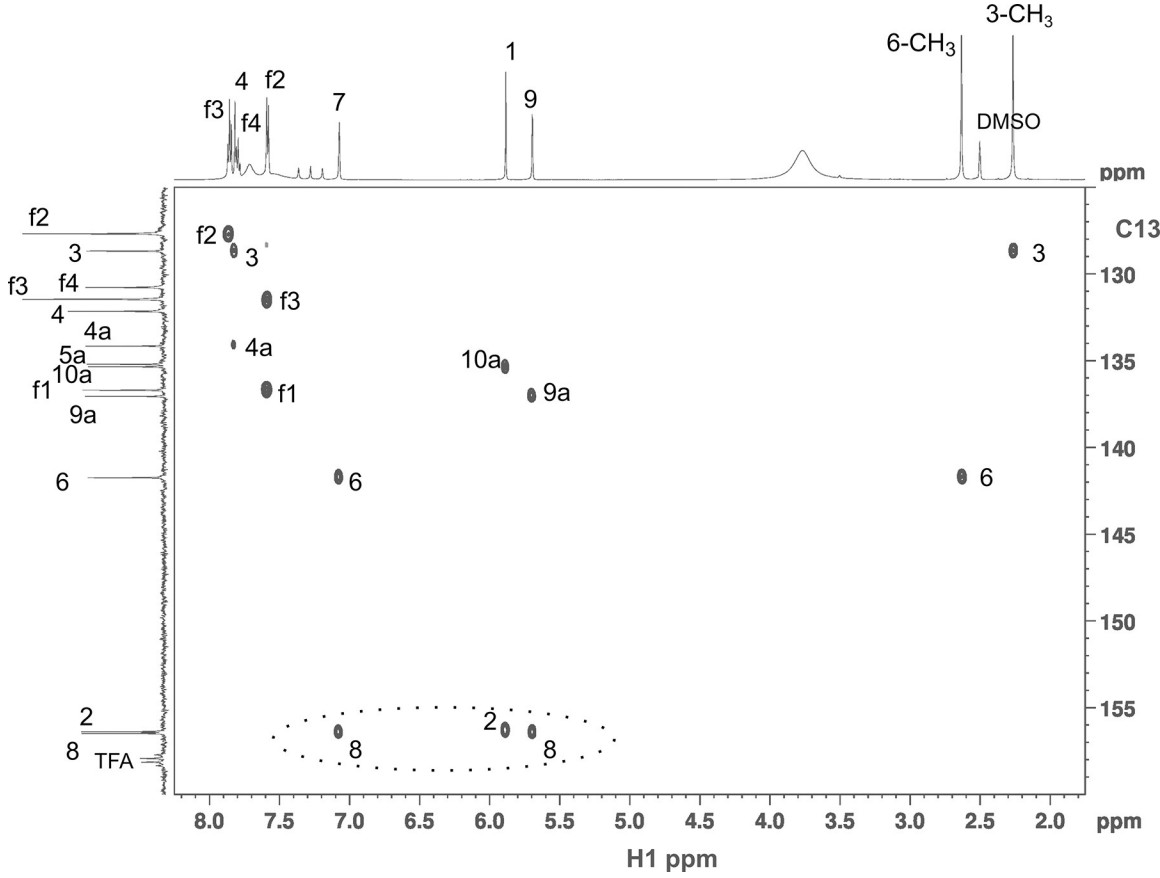

**Fig 5. The 1,1 ADEQUATE spectrum of safranine III.** The boxed area at the lower left is unambiguous evidence that the methyl group in III is at position 6.

## Spectral assignments

**Safranine I.** [1]H-NMR (DMSO, 600 MHz): δ 1.33 [3H, s, C(9)C$H_3$], 2.28 [3H, s, C(3)C$H_3$], 6.07 [1H, s, $H$(1)], 7.33 [1H, d, $J$ = 9.1 Hz, $H$(7)], 7.67 [2H, m, $H$(f2)], 7.77 [3H, m, $H$(f3) & $H$(f4)], 7.82 [1H, s, $H$(4)], 7.86 [1H, d, $J$ = 9.1 Hz, $H$(6)]. [13]C-NMR (DMSO, 150 MHz): δ 13.23 [$C$(9)C$H_3$], 17.05 [$C$(3)C$H_3$], 94.06 [$C$(1)], 101.15 [$C$(9)], 121.00 [$C$(7)], 129.06 [$C$(3)], 129.13 [$C$(f2)], 130.28 [$C$(f3)], 130.42 [$C$(f4)], 131.95 [$C$(4)], 132.33 [$C$(6)], 134.44 [$C$(9a)], 135.00 [$C$(4a)], 136.45 [$C$(5a)], 136.74 [$C$(10a)], 140.07 [$C$(f1)], 156.7[$C$(2)], 157.15 [$C$(8)]. [15]N-NMR (DMSO, 60 MHz, via [15]N-HMBC): δ 97.2 [$C$(2)N$H_2$], 98.0 [$C$(8)N$H_2$ (16)], 160.1 [$N$(10)], 339.3 [$N$(5)].

**Safranine II.** [1]H-NMR (DMSO, 600 MHz): δ 2.26 [6H, s, C(3/7)C$H_3$:($H$(17))], 5.93 [2H, s, $H$(1/9)], 7.61 [2H, d, $J$ = 7.8 Hz, $H$(f2)], 7.75 [2H, s, $H$(4/6)], 7.81 [1H, t, $J$ = 7.5 Hz, $H$(f4)], 7.87 [2H, t, $J$ = 7.5 Hz, $H$(f3)]. [13]C-NMR (DMSO, 150 MHz): d 17.24 [$C$(3/7)C$H_3$], 93.52 [$C$(1/9)], 127.66 [$C$(f2)], 129.00 [$C$(3/7)], 130.80 [$C$(f4)], 131.45 [$C$(f3)], 131.73 [$C$(4/6)], 135.20 [$C$(9a/10a)], 135.47 [$C$(4a/5a)], 136.50 [$C$(f1)], 156.22 [$C$(2/8)]. [15]N-NMR (DMSO, 60 MHz, via [15]N-HMBC): d 95.2 [$C$(2/8)N$H_2$], 159.7 [$N$(10)], 336.8 [$N$(5)].

**Safranine III.** [1]H-NMR (DMSO, 600 MHz): d 2.26 [3H, s, C(3)C$H_3$], 2.63 [3H, s, C(6)C$H_3$], 5.69 [1H, d, $J$ = 2.0 Hz, $H$(9)], 5.88 [1H, s, $H$(1)], 7.07 [1H, bs, $H$(7)], 7.58 [2H, d, $J$ = 7.9 Hz, $H$(f2)], 7.79 [1H, t, $J$ = 7.6 Hz, $H$(f4)], 7.82 [1H, s, $H$(4)], 7.86 [2H, t, $J$ = 7.9 Hz, $H$(f3)]. [13]C-NMR (DMSO, 150 MHz): d 17.15 [$C$(3)C$H_3$], 17.33 [$C$(6)C$H_3$], 92.06 [$C$(1)], 120.06 [$C$

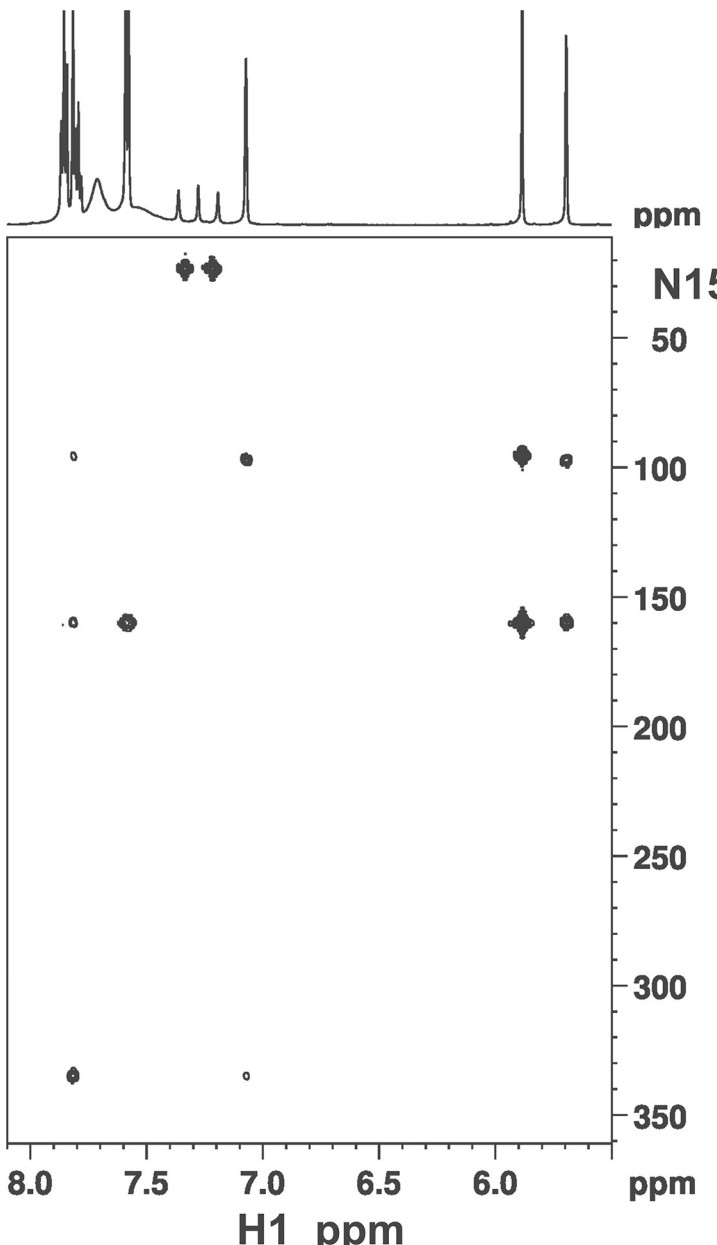

**Fig 6. Natural abundance [15]N-HMBC spectrum of safranine III.** At the top is the correlating 1-dimensional [1]H spectrum. The weak crosspeaks between $H(4)$—$N(10)$ and $H(7)$–$N(5)$ result from 4-bond J-couplings in this highly conjugated system.

(7)], 127.66 [$C$(f2)], 128.66 [$C$(3)], 130.76 [$C$(f4)], 131.44 [$C$(f3)], 132.14 [$C$(4)], 134.14 [$C$(4a)], 135.19 [$C$(5a)], 135.33 [$C$(10a)], 136.68 [$C$(f1)], 137.04 [$C$(9a)], 141.73 [$C$(6)], 156.37 [$C$(2)], 156.48 [$C$(8)]. [15]N-NMR (DMSO, 60 MHz, via [15]N-HMBC): ð 95.3 [$C$(2)$NH_2$], 97.3 [$C$(8) $NH_2$], 159.6 [$N$(10)], 334.6 [$N$(5)].

## Safranine reduction potentials

Fig 7 depicts the potentiometric titration of each purified compound, revealing that safranines **II** and **III** have similar reduction potentials (−272 and −276 mV vs SHE respectively), while

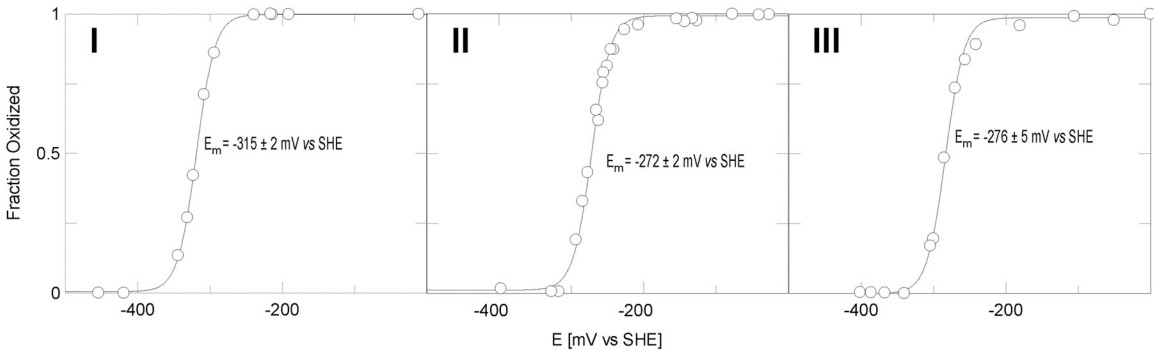

**Fig 7. Potentiometric titration of safranines I-III.** Line drawn is a fit to Eq 1 with the number of electrons fixed at 2. Errors are errors of the fit.

safranine **I** has an approximately 40 mV more negative potential, at −315 mV vs. SHE, than **II** and **III**.

In general, a methyl group in a conjugated system contributes electron density through an inductive effect that is unaffected by solution pH [34]. The Hammett equation describes the thermodynamic equilibrium effects of substituents on aromatic rings for different kinds of reactions where a reaction center is attached to the aromatic ring [35]. It has been shown that the Hammett equation pertains to reduction potentials as well [36]. It is formulated as $log\frac{K}{K_0} = \sigma \times \rho$ where $\frac{K}{K_0}$ describes the ratio of the equilibrium constant of a reaction; $K$, vs. a reference reaction; $K_0$. $\sigma$ is dependent on the position and identity of the substituent and $\rho$ is a reaction specific constant. $\rho$ is positive for reactions where negative charge is built up, or where positive charge disappears which is the case when Safranines are reduced. The Hammett constant, $\sigma$, is negative for methyl groups which means that the product of $\rho$ and $\sigma$ is a negative number. This means that the general effect of methyl substituents that are part of the conjugated aromatic system should be to stabilize the oxidized state of the Safranines, lowering their reduction potentials.

**I** (see Table 1 for the compound numbering scheme) has a methyl substituent with the closest proximity to the positively charged nitrogen of the oxidized Safranine with its two-bond distance as well as being the closest spatially. In **II** the distance is four bonds, and in **III** it is three bonds. It is arguable that the methyl groups stabilize the positive charge both through its electron donating "Hammett" effect as well as its electric field effect since the inductive effect is bond-distance dependent as well as through-space dependent (the strength of the through-space effect being inversely related to the dielectric strength of the solvent) [37]. In fact, the redox potential follows the rank order of the bond distance from the methyl group so this may also explain why the redox potential of **II** is slightly lower than **III** as well. In previous work we have shown that the N(5) nitrogen chemical shift in both safranine analogues altered at the para position of the N(10) phenyl ring [23] and in the N(5) chemical shift of the related flavin molecule in various hydrogen bonding complexes [38, 39] correlates closely with the molecular reduction potential. In this case there is no such correlation. This may be a result of the fact that the methyl groups whose positions change are directly attached to the central phenazine moiety and thus have a much greater inductive effect on the N(5) electron density.

## Conclusions

We have identified the three major components in the commercial Safranine O dye and determined their structures using NMR. In all these compounds the amino groups remain at the

same positions (on C2 and C8) on the opposite sides of the fused ring system whereas the methyl groups to seem to scramble on the carbons of the outer rings. The mechanism for methyl scrambling is not yet known. It seems likely that these other safranine isomers are side products in synthesis as opposed to breakdown products—samples in amber-colored NMR tube are stable for months and we do observe any changes for samples stored in clear tubes and exposed to sunlight for a similar period of time either. These compounds can only be isolated by reversed-phase HPLC: silica gel chromatography does not separate these isomers at all, and in our hands the mixture has been resistant to all attempts at crystallization. Furthermore the presence of the commonly used negative counterion trifluoracetate in the mobile phase makes the safranines co-elute even on the C18 column, probably because they act as the counterion of the positively charged dyes. We have further interpreted the results of potentiometric titrations and have argued that the differences in their potential values can be explained by structural induction effects from the position of the methyl substituents on the phenylphenazinium ring.

Proportions of these safranin compounds may differ depending on vendor formulation which would cause shifts in the apparent and average midpoint potentials of Safranine O. This could be especially problematic when Safranine O is used as a potential standard, as has been the case in a great number of experimental measurements of mitochondrial membrane potentials using safranine fluorescence and absorbance [40]. On the other hand there are also advantages to using a mixture of safranines with a range of midpoint potentials in potentiometric titrations because they establish poise over a greater range of potentials compared to any one pure compound.

## Supporting information

**S1 Data.**
(XLSX)

**S1 File.**
(PDF)

## Author Contributions

**Conceptualization:** Eskil M. E. Andersen, Hsin Wang, Jose F. Cerda, Ronald L. Koder.

**Data curation:** Eskil M. E. Andersen, Hsin Wang, Joshua S. H. Khoo, Jose F. Cerda.

**Formal analysis:** Hsin Wang.

**Funding acquisition:** Ronald L. Koder.

**Investigation:** Eskil M. E. Andersen, Hsin Wang, Joshua S. H. Khoo, Jose F. Cerda.

**Methodology:** Hsin Wang, Jose F. Cerda, Ronald L. Koder.

**Supervision:** Ronald L. Koder.

**Writing – original draft:** Eskil M. E. Andersen, Hsin Wang.

**Writing – review & editing:** Ronald L. Koder.

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
