## [Decision Letter · Decision Letter 0]

7 Apr 2022

PONE-D-22-05311Oxidation-reduction and photophysical properties of isomeric forms of SafraninPLOS ONE

Dear Dr. Koder,

Thank you for submitting your manuscript to PLOS ONE. After careful consideration, we feel that it has merit but does not fully meet PLOS ONE’s publication criteria as it currently stands. Therefore, we invite you to submit a revised version of the manuscript that addresses the points raised during the review process.

We look forward to receiving your revised manuscript.

Kind regards,

Robert A. Niederman, Ph.D

Academic Editor

PLOS ONE

Journal Requirements:

"We gratefully acknowledge funding from NYSERDA grant NYS115213."

We note that you have provided funding information. However, funding information should not appear in the Acknowledgments section or other areas of your manuscript. We will only publish funding information present in the Funding Statement section of the online submission form. 

"This work was supported by a Grant from the New York State Energy Research and Development Authority (NYSERDA grant #NYS115213 to RLK). URL: https://www.nyserda.ny.gov/ The funders had no role in study design, data collection and analysis, decision to publish, or preparation of the manuscript."

Additional Editor Comments:

Editor’s comments Ms. # PONE-D-22-05311

Corrections needed:

Table 1 -- 2,8-diamino-3,7-dimethyl-10

Figs. as displayed in Ms. copy that I received did not appear to be of a quality to reproduce sufficiently well in the final publication.

p. 20, L10: “why the redox potential of II is slightly lower than II as well.” Needs to be clarified??

Reviewers' comments:

Reviewer's Responses to Questions

**Comments to the Author**

1. Is the manuscript technically sound, and do the data support the conclusions?

Reviewer #1: Yes

Reviewer #2: Yes

2. Has the statistical analysis been performed appropriately and rigorously? 

Reviewer #1: N/A

Reviewer #2: N/A

3. Have the authors made all data underlying the findings in their manuscript fully available?

Reviewer #1: Yes

Reviewer #2: Yes

4. Is the manuscript presented in an intelligible fashion and written in standard English?

Reviewer #1: Yes

Reviewer #2: Yes

5. Review Comments to the Author

Reviewer #1: This manuscript is clearly written and describes the careful characterization of commercially prepared Safranine O, which is commonly used as a mediator in electrochemistry. Specifically, they note that during synthesis, two distinct molecules are produced with unique spectral and electrochemical properties. Given that these compounds are widely used and the mixture of compounds complicates their application, it is important that this work be published.

Experiments are well described and the data is clearly presented. Only one minor issue - a relict from a previous submission:

"The reduction potential of Safranine O at pH 7.0 was first reported by Prince et al. *in this journal*

in 1981 to be −280 mV vs the Standard Hydrogen Electrode (SHE) as determined by polarography [9]"

Reviewer #2: This manuscript by Andersen et al describes an analysis of two commercial preparations of safranin, revealing it to be made up of a number of isomers with varying optical and redox properties. The findings will be of interest to users of safranine as a mediator or dye.

The research has been conducted to a high standard and the conclusions drawn seem reasonable. The manuscript is mostly well written and clear, but would benefit from a thorough proof-read.

It would have been helpful for the authors to have included page numbers in their manuscript. In the absence of these the following refers to page numbers in the pdf document

Page 8, second line from bottom – random use of italics

Page 9, line 9 – the authors write “in this journal in 1981”. Does this betray a previous, unsuccessful submission to BBA?

Page 14, Figure legend. Some additional punctuation would be helpful here.

Page 14, last line and Page 15, first line - what does “Figure 3c and 3d” refer to here?

Page 21. All references should be in a consistent format.

6. PLOS authors have the option to publish the peer review history of their article (what does this mean?). If published, this will include your full peer review and any attached files.

Reviewer #1: No

Reviewer #2: No

---

## [Author Response · Author response to Decision Letter 0]

12 May 2022

Dear Dr. Niederman,

Please find attached a re-revised version of our manuscript “Oxidation-reduction and photophysical properties of isomeric forms of Safranin”, PONE-D-22-05311. We have incorporated changes that address all the comments brought forth by both reviewers, and we believe the paper is greatly improved as a result of these suggestions. We thank the reviewers first for these suggestions, and second for responding so quickly. 

A detailed list of the reviewer comments and our responses follows:

Journal Requirements:

(1) Please ensure that your manuscript meets PLOS ONE's style requirements.

We have reformatted the manuscript to fit the Journal’s style guide.

(2) However, funding information should not appear in the Acknowledgments section or other areas of your manuscript. We will only publish funding information present in the Funding Statement section of the online submission form. Please remove any funding-related text from the manuscript and let us know how you would like to update your Funding Statement.

We have removed the Acknowledgements section of the text completely. However, we were recently reminded that we should also acknowledge the funding that supported the NMR spectrometers used in the paper. As such, the we have added NIH grants S10OD018509, S10OD016432, and P41GM066354.

(3) In your Data Availability statement, you have not specified where the minimal data set underlying the results described in your manuscript can be found.

The minimal data set, including NMR data and potentiometric titration data, has been uploaded as supplemental information. 

Reviewer 1:

(1) Only one minor issue - a relict from a previous submission: "The reduction potential of Safranine O at pH 7.0 was first reported by Prince et al. *in this journal* in 1981 to be −280 mV vs the Standard Hydrogen Electrode (SHE) as determined by polarography [9]"

The offending verbiage has been removed. The sentence now reads: “The reduction potential of Safranine O at pH 7.0 was first reported by Prince et al. in 1981 to be −280 mV vs the Standard Hydrogen Electrode (SHE) as determined by polarography [9].” We thank the reviewer for catching our embarrassing error. 

Reviewer 2:

(1) Page 8, second line from bottom – random use of italics.

If we’ve found the correct area, these italics are meant to indicate the atom type being discussed in NMR assignments. This is a common usage in the organic literature.

(2) Page 9, line 9 – the authors write “in this journal in 1981”. Does this betray a previous, unsuccessful submission to BBA?

The offending verbiage has been removed. The sentence now reads: “The reduction potential of Safranine O at pH 7.0 was first reported by Prince et al. in 1981 to be −280 mV vs the Standard Hydrogen Electrode (SHE) as determined by polarography [9].” We thank the reviewer for catching our embarrassing error. 

(3) Page 14, Figure legend. Some additional punctuation would be helpful here.

A number of missing periods were added. We thank the reviewer for catching this.

(4) Page 14, last line and Page 15, first line - what does “Figure 3c and 3d” refer to here?

This verbiage was left over from a previous version of the figure. The sentence now reads: “The major HPLC component, II, displays 1H and 13C-spectra characteristic of a molecule with C2 symmetry (Figure 3), and can be readily identified as authentic safranine O.” We thank the reviewer for catching this.

Alterations in the manuscript are highlighted in yellow. We thank you for your time and attention, and hope these changes make a convincing argument for publication.

Best Regards,

Ronald L. Koder, Ph.D.

---

## [Editor Report · Decision Letter 1]

3 Jun 2022

Oxidation-reduction and photophysical properties of isomeric forms of Safranin

PONE-D-22-05311R1

Dear Dr. Koder,

We’re pleased to inform you that your manuscript has been judged scientifically suitable for publication and will be formally accepted for publication once it meets all outstanding technical requirements. Please note that an error on p.13, line 12 of the revised Ms. was found (should read: why the redox potential of II is slightly lower than III as well.) that can be corrected in the galley proofs.

Kind regards,

Robert A. Niederman, Ph.D

Academic Editor

PLOS ONE

Additional Editor Comments (optional):

Please note that an error on p.13, line 12 of the revised Ms. was found (should read: why the redox potential of II is slightly lower than III as well.) that can be corrected in the galley proofs.

---

## [Editor Report · Acceptance letter]

16 Jun 2022

PONE-D-22-05311R1 

Oxidation-reduction and photophysical properties of isomeric forms of Safranin 

Dear Dr. Koder:

I'm pleased to inform you that your manuscript has been deemed suitable for publication in PLOS ONE. Congratulations! Your manuscript is now with our production department. 

Kind regards, 

on behalf of

Dr. Robert A. Niederman 

Academic Editor

PLOS ONE